# Model-Based Non-Independent Distortion Cost Design for JPEG Steganography with Symmetric Embeeding

## ABSTRACT

Recent achievements have shown that model-based steganographic schemes hold promise for better security than heuristic-based ones, as they can provide theoretical guarantees on secure steganography under a given statistical model. However, it remains a challenge to exploit the correlations between DCT coefficients for secure steganography in practical scenarios where only a single compressed JPEG image is available. To cope with this, we propose a novel model-based steganographic scheme using the Conditional Random Field (CRF) model with four-element cross-neighborhood to capture the dependencies among DCT coefficients for JPEG steganography with symmetric embedding. Specifically, the proposed CRF model is characterized by the delicately designed energy function, which is defined as the weighted sum of a series of unary and pairwise potentials, where the potentials associated with the statistical detectability of steganography are formulated as the KL divergence between the statistical distributions of cover and stego. By optimizing the constructed energy function with the given payload constraint, the non-independent distortion cost corresponding to the least detectability can be accordingly obtained. Extensive experimental results validate the effectiveness of our proposed scheme, especially outperforming the previous independent art J-MiPOD.

## CCS CONCEPTS

• **Security and privacy** → **Security services**; **Social aspects of security and privacy**.

## KEYWORDS

JPEG Image Steganography, Gaussian Markov Random Field (GMRF), KL-divergence, statistical modeling

## 1 INTRODUCTION

Digital steganography [7] is an important branch of information hiding that utilizes the data redundancy inherent in various types of digital media, as well as the physiological and psychological characteristics of human perception organs, to embed secret messages (a.k.a., payload) into the public digital media with certain coding, and then transmit the digital media containing the secret messages to achieve covert communication. As opposed to traditional encryption, steganography is concerned with concealing the communication behavior, making the communication process imperceptible or even undetectable.

As one of the most commonly used digital media, digital image has always been the popular cover choice for steganography, and has yielded numerous impressive results in recent years [26, 31]. Leaving aside the current new paradigm of coverless generative steganography, the content-adaptive steganography based on the distortion minimization framework [5] has been the mainstream steganographic paradigm. And with the advent of Syndrome-Trellis Codes (STCs) that can closely achieve the rate-distortion bound [6], the main effort in content-adaptive image steganography has focused on designing efficient distortion cost functions. To ensure the security of covert communication, it is required that the distortion cost function be designed to least possibly change the statistical distributions of the cover source for a given payload.

For the time being, the design of distortion cost functions can be divided into three categories, i.e., heuristic-based, deep learning (DL)-based, and model-based. The first type relies on heuristic principles, whose guideline is to assign high costs to smooth areas and low costs to textured regions, based on the consensus that textured regions are difficult to model while smooth areas are easy, typically schemes are WOW [14], S-UNIWARD [16], HILL [22] in the spatial domain and UERD [13], GUED [30], J-UNIWARD [16], DCDT [29] in the JPEG domain. Despite the impressive achievements of these empirical schemes, they are limited from the methodological point of view, since they can not provide accurate explanations as to why the given cost functions work. The second one, DL-based approaches, which have achieved excellent security performance, are generally designed as an automatic cost learning framework via utilizing the architectures of generative adversarial networks and reinforcement learning, typical schemes are UT-GAN [39], JS-GAN [40], SPAR-RL [36], JEC-RL [37], PICO-RL [23], Steg-GMAN [18]. These methods use dynamic feedback from the discriminator (i.e., steganalyzer) to iteratively facilitate the generator to output better distortion cost to defeat the given "adversary". However, such methods are end-to-end and lack mathematical explanation as well. In addition, they rely heavily on well-designed networks and require both parties to form a Nash equilibrium. What's more, they are computationally demanding and data-dependent. The last type is the model-based, which builds the steganographic distortion cost based on mathematical principles, including minimizing the difference of model distribution between cover and stego (e.g., MG [8], MVGG [25], GMRF [28]), and minimizing the power of optimal detector (e.g., MiPOD [24], J-MiPOD [3]). Not only does this type of scheme sound more mathematical, but they can provide a complete mathematical explanation, and for this reason, this paper will continue to explore such type in depth.

MG [8] is the first practical attempt to design the distortion function based on the mathematical principle. The distortion costs are determined with the embedding change probabilities, which are

*ACM MM, 2024, Melbourne, Australia*
© 2024 Copyright held by the owner/author(s). Publication rights licensed to ACM.
ACM ISBN 978-x-xxxx-xxxx-x/YY/MM
https://doi.org/10.1145/nnnnnnn.nnnnnnn

obtained by minimizing the Kullback-Leibler (KL) divergence between the statistical distributions of cover and stego images, when the cover image is modeled as a sequence of independent multivariate Gaussian random variables with heterogeneous local variances. And the security performance is further improved in MVGG [25] by incorporating an improved variance estimator and adopting the pentanary symmetric embedding with a thicker-tail model. Developing upon these two works, an alternative approach, i.e., Minimizing the Power of Optimal Detector (MiPOD) [24], has been proposed subsequently, with a performance close to the state-of-the-art compared to S-UNIWARD and HILL. In MiPOD, a closed-form expression for statistical detectability is also provided, allowing the design of so-called detectability-limited sender, i.e., controls the size of secure payload for a given image to not exceed a target detectability level. As for the acquisition of the embedding change probabilities associated with the distortion costs, it is derived by minimizing the power of the optimal likelihood ratio test (LRT). In light of recent research [9, 33] showing that DCT coefficients can be modeled individually, Cogranne *et al.* follow MiPOD and build a heteroscedastic model for the DCT coefficients, i.e., modeling DCT coefficients as independent but not identically distributed multivariate Gaussians. This extension of MiPOD for the JPEG images, which we referred to as J-MiPOD [3], is also competitive as compared to the state-of-the-art J-UNIWARD.

Despite the theoretical guarantees of security of the model-based MiPOD and J-MiPOD, their assumption of independence of the pixels/DCT coefficients is not very accurate. In this regard, some works have focused on developing more accurate possible statistical models of images by taking the correlations between pixels/DCT coefficients carefully into account. For example, [28] introduces a Gaussian Markov Random Field (GMRF) model with four-element cross-neighborhood to capture the dependencies among spatially adjacent (i.e., horizontal and vertical) pixels, and attains the embedding change probability for each pixel by minimizing the KL divergence in terms of a series of pairwise cliques between cover and stego within the GMRF model, which achieves superior security performance than the previous independent MiPOD. As for the exploration of the correlations between DCT coefficients for JPEG steganography, some recent works [11, 12] based on the complete and exact knowledge of the acquisition and processing pipeline (a.k.a., side-information in RAW image) build more accurate statistical models for JPEG images. Although it can provide significant performance improvements, it is too conditionally demanding and often difficult to achieve in practice. Therefore, giving solutions in the broader context of not providing side-information of images and using only a single compressed JPEG image deserves further investigation.

To address this issue, in this paper, we propose a novel model-based steganographic scheme, using the pairwise Conditional Random Field (CRF) model with four-element cross-neighborhood to capture the dependencies among DCT coefficients for JPEG steganography, denoted as CRF. Note that for the sake of simplicity, similarly to the previous art [28], only ternary symmetric embedding is investigated in this paper, and the impact brought by the direction of embedding modification is not considered for the time being. Different from the construction of the previous art [28], our

proposed CRF model is not built on the spatially adjacent DCT coefficients, but rather on the same DCT mode in adjacent DCT blocks, since the correlation between spatially adjacent DCT coefficients inside one DCT block can hardly be obtained for a single compressed JPEG image. The proposed CRF model with four-element cross-neighborhood is characterized by the delicately designed energy function, which is defined as the weighted sum of a series of unary and pairwise potentials since the effects of each unary and pairwise potential on steganographic security are different. In our proposed scheme, the unary and pairwise potentials associated with the statistical detectability of steganography are defined as the KL divergence between the statistical distributions of cover and stego. With the aid of the proposed CRF model, secure JPEG image steganography is finally formulated as the minimization of the energy function, and the optimal embedding change probabilities of each DCT coefficient are attained by minimizing the total potentials (i.e., KL divergence) under the payload constraint. With the given embedding change probability, the corresponding steganographic distortion cost can be easily determined. Extensive experiments are carried out to verify the effectiveness of the proposed scheme (known as CRF) using CC-JRM [19], DCTR [15], GFR [27] and SCA-GFR [4], on BOSSbase [1] and ALASKAv2 [2] database. Numerous results show that the proposed method can not only surpass the SOAT heuristic-based UERD and JUNIWARD, but also outperform the advanced model-based J-MiPOD by a clear margin.

## 2 PRELIMINARIES

### 2.1 Notations and Basic Concepts

In this paper, matrixes and vectors are represented by capital and lowercase boldface symbols, respectively, sets are denoted by the calligraphic font, and elements within a matrix are indicated by italic fonts with subscript indices. $\mathbf{A}^{\mathrm{T}}$ is the transpose of matrix $\mathbf{A}$, and Pr is the probability measure.

For ease of presentation, in this paper, the JPEG gray-scale cover and stego images with size $h \times w$ are denoted as $\mathbf{X} = \left(x_{m,n}^{k,l}\right)^{h \times w}$ and $\mathbf{Y} = \left(y_{m,n}^{k,l}\right)^{h \times w}$, respectively, where $1 \leq m \leq h/8$, $1 \leq n \leq w/8$, $0 \leq k, l \leq 7$, $h$ and $w$ are taken the integer multiples of 8. $x_{m,n}^{k,l} \in \mathcal{I} = \{-1024, -1023, \ldots, 1023\}$ (or $y_{m,n}^{k,l}$) indicates the DCT coefficient with the $(k, l)$-th DCT frequency mode in the $(m, n)$-th DCT block of $\mathbf{X}$ (or $\mathbf{Y}$), $\beta_{m,n}^{k,l}$ is the corresponding embedding change probability. The DCT basis for the $(k, l)$-th frequency mode is defined as an $8 \times 8$ matrix $\mathbf{f}^{k,l} = \left(f_{i,j}^{k,l}\right)^{8 \times 8}$, where

$$f_{i,j}^{k,l} = \frac{w_k w_l}{4} \cos \frac{\pi k(2i + 1)}{16} \cos \frac{\pi l(2j + 1)}{16}, \qquad (1)$$

$w_0 = 1/\sqrt{2}$, $w_k = 1$ for $k > 0$, $i$ and $j$ are the corresponding spatial indexes inside a $8 \times 8$ block. By performing Inverse Discrete Cosine Transform on the DCT coefficients in $\mathbf{X}$, we can obtain the corresponding decompressed spatial image $\mathbf{Z} = \left(z_{m,n}^{i,j}\right)^{h \times w}$, where

$$z_{m,n}^{i,j} = \sum_{k=0}^{7} \sum_{l=0}^{7} f_{i,j}^{k,l} q_{k,l} x_{m,n}^{k,l}, \qquad (2)$$

$q_{k,l}$ is the $(k,l)$-th quantization step in the JPEG luminance quantization matrix.

## 2.2 The CRF Model for JPEG Steganography

As stated above, this paper focuses on the broader context where there is only a single JPEG image without any side-information about the cover source. In exploring the correlations between DCT coefficients for JPEG steganography, we will only consider the correlations of DCT coefficients from the same mode in horizontally or vertically adjacent blocks, since its correlations are the most significant, as evidenced by [35]. In this way, as illustrated in Figure 1 ($B_{m,n}$ is the $(m,n)$-th $8 \times 8$ DCT block), the JPEG image with size $h \times w$ can be split into 64 sub-images with size $\frac{h}{8} \times \frac{w}{8}$, depending on the DCT modes. Due to the formulation of considered correlations being identically constituted in each sub-image, **the derivation will be performed in this paper using only one arbitrary sub-image for brevity**.

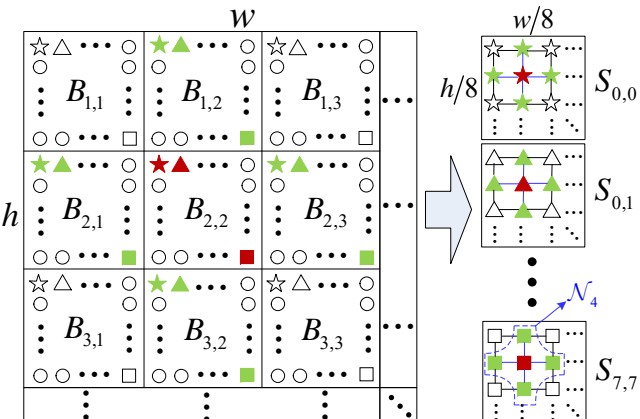

**Figure 1: The illustration of splitting a JPEG image depending on DCT modes. The DCT coefficients with the same mode are regrouped into the same sub-image.**

Without loss of generality, we denote the $(k,l)$-th sub-image by S, omitting the index $(k,l)$ in the subsequent theoretical derivation for the sake of simplicity and readability. Specifically, to incorporate the considered correlations, we first define a Conditional Random Field (S, $\boldsymbol{\beta}$) on an undirected graph $\mathcal{S} = (\mathcal{V}, \mathcal{E})$. S is the observed sub-image, $\boldsymbol{\beta}$ is a set of embedding change probabilities, and each $\beta_{m,n} \in \boldsymbol{\beta}$ corresponds to the one of $(m,n)$-th DCT coefficient $x_{m,n}$ in S. As shown in Figure 1, $\mathcal{S}$ consists of numerous nodes and edges, wherein $\mathcal{V} = \{(m,n)|1 \le m \le h/8, 1 \le n \le w/8\}$ is a set that collects the node index for the grid-like structure of DCT coefficients, $\mathcal{E}$ is a set that collects the edges connecting horizontally or vertically adjacent nodes. Since only the correlations of horizontally or vertically adjacent nodes are considered in this paper, thus we will use the four-element cross-neighborhood $\mathcal{N}_4$ (see Figure 1) to characterize $\mathcal{S}$. Inside this four-element cross-neighborhood system, there are two types of cliques, i.e., one-element (unary) cliques and two-element (pairwise) cliques. Formally, according to the Hammersley Clifford theorem [21], the posterior distribution $\Pr(\boldsymbol{\beta} | S)$ of any

possible assignment to $\boldsymbol{\beta}$ conditioned on S is a Gibbs distribution and defined as:

$$\Pr(\boldsymbol{\beta} | S) = \frac{1}{\Lambda} \exp\left(-\sum_{c \in C} \psi_c(\boldsymbol{\beta}_c | S)\right), \qquad (3)$$

where $C$ is the set of all unary and pairwise cliques in S, $\psi_c(\boldsymbol{\beta}_c | S)$ is the potential function defined on clique $c$, and $\Lambda$ is a normalizing constant known as the partition function. In a CRF model, the corresponding energy function $E(\boldsymbol{\beta} | S)$ is defined as:

$$E(\boldsymbol{\beta} | S) = -\log \Pr(\boldsymbol{\beta} | S) - \log \Lambda = \sum_{c \in C} \psi_c(\boldsymbol{\beta}_c | S), \qquad (4)$$

and the optimal assignment $\boldsymbol{\beta}^*$, i.e. the maximum posterior (MAP) probability of the random field **B**, is given by:

$$\boldsymbol{\beta}^* = \arg\max_{\boldsymbol{\beta}} \Pr(\boldsymbol{\beta} | S) = \arg\min_{\boldsymbol{\beta}} E(\boldsymbol{\beta} | S). \qquad (5)$$

It manifests that the problem of finding the optimal embedding probabilities for secure JPEG image steganography can be formulated as the minimization of a well-designed energy function.

Concretely, the energy function in Eq. (4) corresponding to the proposed CRF model with four-element cross-neighborhood can be formulated as the sum of two kinds of local potentials $\varphi_{m,n}$ and $\psi_c$:

$$E(\boldsymbol{\beta} | S) = \sum_{(m,n) \in \mathcal{V}} \varphi_{m,n}(\beta_{m,n} | x_{m,n}) + \sum_{c \in \mathcal{E}} \psi_c(\boldsymbol{\beta}_c | \boldsymbol{x}_c). \qquad (6)$$

Herein, $\varphi_{m,n}$ is referred to as the unitary potential, employed to characterize the detectability of embedding messages in individual DCT coefficient $x_{m,n}$ in $\mathcal{V}$ with embedding change probability $\beta_{m,n}$, and $\psi_c$ is called the pairwise potential, used to characterize the detectability of embedding messages in the correlated DCT coefficient pair $\boldsymbol{x}_c$ in $\mathcal{E}$ with embedding change probabilities $\boldsymbol{\beta}_c$. In this way, maximizing the security of JPEG image steganography is then formulated as minimizing the potentials in Eq. (6).

As for the characterization of steganographic detectability, there is a well-recognized metric, supported by a rigorous mathematical theory, namely KL divergence, which has already been used in recent works [8, 10, 28]. It shows that the smaller the KL divergence between the statistical distributions of cover and stego, the higher the security of steganography, which is exactly consistent with the relations between potentials in Eq. (6) and steganographic security. Based on this observation, in this paper, we will formulate both the unary and pairwise potentials in Eq. (6) as KL divergence between the statistical distributions of cover and stego.

## 3 THE PROPOSED JPEG STEGANOGRAPHIC DISTORTION COST DESIGN USING CRF MODEL

### 3.1 Overview

For considering the correlations between DCT coefficients for JPEG steganography with symmetric embedding, we propose a Conditional Random Field (CRF) model with four-element cross-neighborhood to capture the correlation between DCT coefficients for non-independent steganographic distortion cost design, which has not been considered in the prior art J-MiPOD [3]. With the aid of the proposed model, the JPEG steganographic distortion cost design is then formulated as minimizing the total potentials (all unary

and pairwise) under the payload constraint. In this section, we first develop the statistical models for both JPEG cover and stego images, which are used to characterize the correlations between DCT coefficients. Next, define the unary and pairwise potentials using KL divergence between the statistical distributions of cover and stego. Finally, formulate the distortion cost design as the minimization of total potentials and provide an iterative optimization approach to obtain the optimal embedding probabilities, associated with distortion cost.

### 3.2 JPEG Cover and Stego Image Models

*3.2.1 Cover model.* As discussed above, a CRF model with four-element cross-neighborhood system is utilized to model the cover images in this paper, and only the correlations between DCT coefficients of the same DCT modes in horizontally and vertically adjacent DCT blocks are considered. For this reason, we develop the statistical model for pairwise DCT coefficients in JPEG cover and stego images. **For simplicity of the exposition and without prejudice to generality, we take a pairwise DCT coefficient $x_c = [x_{m,n}, x_{m,n+1}]$ in S as an instance in the subsequent derivation.**

As stated in [12, 32, 34, 35], the DCT coefficients in a JPEG image, which are typically corrupted by Gaussian noise developed from the RAW image acquisition up to the JPEG compression, are feasible to be modeled as multivariate Gaussian random variables. Based on this insight, we move one step forward by modeling pairwise coefficients $x_c$ in a single compressed JPEG image as a bivariate Gaussian random variable quantized to integer. Under the assumption of fine quantization limit (refer to [8, 25, 28]), the Probability Mass Function (PMF) of $x_c$ is defined as $\mathcal{P}_{\Sigma_c} = (p(t_1, t_2))_{t_1, t_2 \in \mathcal{I}}$ with:

$$p(t_1, t_2) = \Pr(x_{m,n} = t_1, \ x_{m,n+1} = t_2)$$
$$\propto \frac{1}{2\pi |\Sigma_c|^{1/2}} \exp\left(-\frac{1}{2}(t_c - \mu_c)\Sigma_c^{-1}(t_c - \mu_c)^{\mathrm{T}}\right), \quad (7)$$

where $t_c = [t_1, t_2]$ and $\mu_c = [\mu_{m,n}, \mu_{m,n+1}]$ are the realization and expectation of $x_c$, respectively. $\Sigma_c$ and $\Sigma_c^{-1}$ are the corresponding $2 \times 2$ covariance matrix and its inverse matrix, respectively:

$$\Sigma_c = \begin{bmatrix} \sigma_{m,n}^2 & \varrho_c \sigma_{m,n} \sigma_{m,n+1} \\ \varrho_c \sigma_{m,n} \sigma_{m,n+1} & \sigma_{m,n+1}^2 \end{bmatrix}, \quad (8)$$

$$\Sigma_c^{-1} = \begin{bmatrix} \dfrac{1}{(1 - \varrho_c^2)\sigma_{m,n}^2} & \dfrac{-\varrho_c}{(1 - \varrho_c^2)\sigma_{m,n}\sigma_{m,n+1}} \\ \dfrac{-\varrho_c}{(1 - \varrho_c^2)\sigma_{m,n}\sigma_{m,n+1}} & \dfrac{1}{(1 - \varrho_c^2)\sigma_{m,n+1}^2} \end{bmatrix}. \quad (9)$$

Here, $\varrho_c$ is the correlation coefficient between $x_{m,n}$ and $x_{m,n+1}$, $\sigma_{m,n}^2$ and $\sigma_{m,n+1}^2$ are the variance of $x_{m,n}$ and $x_{m,n+1}$, respectively. Note that, all these parameters have to be estimated from a single JPEG image, which will be described later in Section IV.

In addition, according to [38], for a multivariate Gaussian random variable, its marginal variables are also Gaussian distributed, i.e., $x_{m,n}$ still follows a Gaussian distribution. Thereby, the PMF of $x_{m,n}$

can be given by $\mathcal{P}_{\sigma_{m,n}} = (p(t_1))_{t_1 \in \mathcal{I}}$ with

$$p(t_1) = \Pr(x_{m,n} = t_1) \propto \frac{1}{\sqrt{2\pi}\sigma_{m,n}} \exp\left(-\frac{(t_1 - \mu_{m,n})^2}{2\sigma_{m,n}^2}\right). \quad (10)$$

*3.2.2 Stego model.* Note that for the sake of brevity, we only consider ternary symmetric embedding in our proposed scheme for the time being, under which the statistical model for the stego image Y can be easily derived from the corresponding cover image model. Specifically, under ternary symmetric embedding, with the given payload constraint, each $x_{m,n}$ in S will be modified by $\{+1, 0, -1\}$ with embedding change probabilities $\{\beta_{m,n}, 1-2\beta_{m,n}, \beta_{m,n}\}$, respectively. Therefore, given a pairwise cover coefficient $x_c = [x_{m,n}, x_{m,n+1}]$ and the relevant embedding change probabilities $\beta_c = [\beta_{m,n}, \beta_{m,n+1}]$, the corresponding pairwise stego coefficient pair $y_c = [y_{m,n}, y_{m,n+1}]$ can be obtained via applying the following probabilistic rules:

$$\begin{aligned} \Pr(y_{m,n} = x_{m,n}, \ y_{m,n+1} = x_{m,n+1}) &= (1-2\beta_{m,n})(1-2\beta_{m,n+1}), \\ \Pr(y_{m,n} = x_{m,n} \pm 1, \ y_{m,n+1} = x_{m,n+1}) &= \beta_{m,n}(1-2\beta_{m,n+1}), \\ \Pr(y_{m,n} = x_{m,n}, \ y_{m,n+1} = x_{m,n+1} \pm 1) &= (1-2\beta_{m,n})\beta_{m,n+1}, \\ \Pr(y_{m,n} = x_{m,n} \pm 1, \ y_{m,n+1} = x_{m,n+1} \pm 1) &= \beta_{m,n}\beta_{m,n+1}. \end{aligned} \quad (11)$$

Then, the PMF of $y_c$, i.e., $Q_{\Sigma_c, \beta_c} = (q_{\beta_c}(t_1, t_2))_{t_1, t_2 \in \mathcal{I}}$ can be accordingly attained with

$$\begin{aligned} q_{\beta_c}(t_1, t_2) &= \Pr(y_{m,n} = t_1, y_{m,n+1} = t_2) \\ &= (1-2\beta_{m,n})(1-2\beta_{m,n+1}) p(t_1, t_2) \\ &\quad + \beta_{m,n}(1-2\beta_{m,n+1})[p(t_1+1, t_2) + p(t_1-1, t_2)] \\ &\quad + (1-2\beta_{m,n})\beta_{m,n+1}[p(t_1, t_2+1) + p(t_1, t_2-1)] \\ &\quad + \beta_{m,n}\beta_{m,n+1}[p(t_1+1, t_2+1) + p(t_1-1, t_2+1)] \\ &\quad + \beta_{m,n}\beta_{m,n+1}[p(t_1+1, t_2-1) + p(t_1-1, t_2-1)]. \end{aligned} \quad (12)$$

Similarly, the PMF of the corresponding stego coefficient $y_{m,n}$, i.e., $\beta_{m,n} = (q_{\beta_{m,n}}(t_1))_{t_1 \in \mathcal{I}}$ can be obtained with

$$\begin{aligned} q_{\beta_{m,n}}(t_1) &= \Pr(y_{m,n} = t_1) \\ &= (1-2\beta_{m,n})p(t_1) + \beta_{m,n}[p(t_1+1) + p(t_1-1)]. \end{aligned} \quad (13)$$

### 3.3 Unary and Pairwise Potential Formulation

*3.3.1 Unary Potential.* As stated in Sec. 2.2, the potential functions are formulated by KL divergence in our proposed scheme, and the statistical distributions $\mathcal{P}_{\sigma_{m,n}}$ and $Q_{\sigma_{m,n}, \beta_{m,n}}$ of cover and stego images associated with the calculation of KL divergence have been obtained, then the unary potential $\varphi_{m,n}(\beta_{m,n} | x_{m,n})$ can be derived by

$$\varphi_{m,n}(\beta_{m,n} | x_{m,n}) = D_{\mathrm{KL}}\left(\mathcal{P}_{\sigma_{m,n}} \| Q_{\sigma_{m,n}, \beta_{m,n}}\right) \approx \frac{1}{2}\beta_{m,n}^2 I_{m,n}, \quad (14)$$

where

$$I_{m,n} = \sum_{t_1} \frac{1}{p(t_1)}\left(\frac{\partial q_{\beta_{m,n}}(t_1)}{\partial \beta_{m,n}}\right)^2 \approx \frac{2}{\sigma_{m,n}^4} \quad (15)$$

is the steganographic Fisher information, indicating the impact of information hiding on the cover model.

The derivation of Eq. (14) is based on the conclusion that for small embedding change probabilities, the KL divergence is well-approximated by its leading quadratic term of Taylor expansion, the detailed derivation process can refer to the prior arts [8, 28].

Eventually, in conjunction with Eq. (14) and Eq. (15), the unary potential $\varphi_{m,n}(\beta_{m,n}\,|\,x_{m,n})$ is formulated as

$$\varphi_{m,n}(\beta_{m,n}\,|\,x_{m,n}) = \frac{\beta_{m,n}^2}{\sigma_{m,n}^4}. \tag{16}$$

3.3.2 *Pairwise Potential.* As for the pairwise potential $\psi_c(\boldsymbol{\beta}_c|\boldsymbol{x}_c)$ in Eq. (6), it plays a prominent part in capturing the correlation between DCT coefficients for the measure of statistical detectability, leading to the non-independent steganographic distortion cost design attain. Similar to the derivation of $\varphi_{m,n}(\beta_{m,n}\,|\,x_{m,n})$, $\psi_c(\boldsymbol{\beta}_c|\boldsymbol{x}_c)$ can be formalized as the KL divergence between the statistical distributions $\mathcal{P}_{\Sigma_c}$ of $\boldsymbol{x}_c$ and $\mathcal{Q}_{\Sigma_c,\boldsymbol{\beta}_c}$ of $\boldsymbol{y}_c$, and obtained by the leading quadratic term of Taylor expansion of the KL divergence, i.e.,

$$\psi_c(\boldsymbol{\beta}_c|\boldsymbol{x}_c) = D_{\mathrm{KL}}\Big(\mathcal{P}_{\Sigma_c}\|\mathcal{Q}_{\Sigma_c,\boldsymbol{\beta}_c}\Big) \approx \frac{1}{2}\boldsymbol{\beta}_c\mathbf{I}_c\boldsymbol{\beta}_c^{\mathrm{T}}, \tag{17}$$

where $\mathbf{I}_c$ is the $2\times 2$ Fisher information matrix (FIM):

$$\mathbf{I}_c = \begin{bmatrix} I_{11}^{(c)} & I_{12}^{(c)} \\ I_{21}^{(c)} & I_{22}^{(c)} \end{bmatrix}, \tag{18}$$

in which

$$I_{11}^{(c)} = \sum_{t_1,t_2} \frac{1}{p(t_1,t_2)}\left(\frac{\partial q_{\boldsymbol{\beta}_c}(t_1,t_2)}{\partial \beta_{m,n}}\right)^2 \approx \frac{2}{\sigma_{m,n}^4\left(1-\varrho_c^2\right)^2}, \tag{19}$$

$$I_{22}^{(c)} = \sum_{t_1,t_2} \frac{1}{p(t_1,t_2)}\left(\frac{\partial q_{\boldsymbol{\beta}_c}(t_1,t_2)}{\partial \beta_{m,n+1}}\right)^2 \approx \frac{2}{\sigma_{m,n+1}^4\left(1-\varrho_c^2\right)^2}, \tag{20}$$

$$I_{12}^{(c)} = I_{21}^{(c)} = \sum_{t_1,t_2} \frac{1}{p(t_1,t_2)}\left(\frac{\partial q_{\boldsymbol{\beta}_c}(t_1,t_2)}{\partial \beta_{m,n}}\right)\left(\frac{\partial q_{\boldsymbol{\beta}_c}(k_1,t_2)}{\partial \beta_{m,n+1}}\right)$$
$$\approx \frac{2\varrho_c^2}{\sigma_{m,n}^2\sigma_{m,n+1}^2\left(1-\varrho_c^2\right)^2}. \tag{21}$$

One can refer to the prior art [28] for the detailed derivations of FIM, which is dedicated to spatial image steganography. Finally, combining Eq. (17) - Eq. (21), the pairwise potential $\psi_c(\boldsymbol{\beta}_c|\boldsymbol{x}_c)$ is formulated as

$$\psi_c(\boldsymbol{\beta}_c|\boldsymbol{x}_c) = \frac{\beta_{m,n}^2}{\sigma_{m,n}^4\left(1-\varrho_c^2\right)^2} + \frac{\beta_{m,n+1}^2}{\sigma_{m,n+1}^4\left(1-\varrho_c^2\right)^2} + \frac{2\varrho_c^2\,\beta_{m,n}\,\beta_{m,n+1}}{\sigma_{m,n}^2\sigma_{m,n+1}^2\left(1-\varrho_c^2\right)^2}. \tag{22}$$

Referring to Eq. (22), it can be seen that the correlation between DCT coefficients $x_{m,n}$ and $x_{m,n+1}$ has been taken into account by the introduction of correlation coefficient $\varrho_c$. In addition, a closer look at Eq. (16) and Eq. (22) reveals that both unary and pairwise potentials are inversely proportional to the variance of DCT coefficient, implying that modifying the DCT coefficients with small variances are more detectable than in those with large variances. On the other hand, we know that the DCT coefficients with small variances are generally located in the smooth regions or high-frequency modes of JPEG images, and these DCT coefficients have been confirmed by previous studies [3, 13, 16, 29, 30] to be really unsuitable for embedding modifications. In this reagrd, we can claim that the construction of potentials in both Eq. (16) and Eq. (22) for secure JPEG steganography is theoretically feasible.

## 3.4 Optimization Problem Formulation and Solving for Distortion Cost Design

Once the detailed expression of unary and pairwise potentials are determined (see Eq. (16) and Eq. (22)), the energy $E(\boldsymbol{\beta}\mid\mathbf{S})$ can be accordingly obtained by summing all the unary and pairwise potentials in the sub-image $\mathbf{S}$. Note that $\mathbf{S}$ is only one of the 64 mutually independent sub-images, then we expand the derived results to the entire image $\mathbf{X}$, and have

$$E(\boldsymbol{\beta}\mid\mathbf{X}) = \sum_{k=0}^{7}\sum_{l=0}^{7} E_{k,l}(\boldsymbol{\beta}\mid\mathbf{S}), \tag{23}$$

where $\boldsymbol{\beta} = \left(\beta_{m,n}^{k,l}\right)^{h\times w}$ are the embedding change probabilities, $E_{k,l}(\boldsymbol{\beta}\mid\mathbf{S})$ is the energy of the $(k,l)$-th subimage $\mathbf{S}$ and rewritten as:

$$E_{k,l}(\boldsymbol{\beta}\mid\mathbf{S}) = \sum_{(m,n)\in\mathcal{V}} \varphi_{m,n}^{k,l}(\beta_{m,n}\,|\,x_{m,n}) + \sum_{c\in\mathcal{E}} \psi_c^{k,l}(\boldsymbol{\beta}_c\,|\,x_c)$$
$$= \sum_{m=1}^{h/8}\sum_{n=1}^{w/8} \frac{1}{2}\left(\beta_{m,n}^{k,l}\right)^2 I_{m,n}^{k,l} + \sum_{c\in\mathcal{E}_{k,l}} \frac{1}{2}\boldsymbol{\beta}_c^{k,l}\mathbf{I}_c^{k,l}\left(\boldsymbol{\beta}_c^{k,l}\right)^{\mathrm{T}}. \tag{24}$$

However, a closer look at Eq. (23) and Eq. (24) reveals that all the DCT coefficients are treated indistinguishably, which is inconsistent with domain knowledge in steganography. Previous arts [17, 29, 30] have shown that image regions with different complexity, as well as different frequency modes, can afford different levels of anti-detectability. Given this, we further modify Eq. (23) and Eq. (24) by incorporating the domain knowledge in steganography as follows:

$$\begin{cases} \hat{E}(\boldsymbol{\beta}\mid\mathbf{S}) = \sum_{k=0}^{7}\sum_{l=0}^{7} w_{k,l}\cdot\hat{E}_{k,l}(\boldsymbol{\beta}\mid\mathbf{S}), \\ \hat{E}_{k,l}(\boldsymbol{\beta}\mid\mathbf{S}) = \sum_{m=1}^{h/8}\sum_{n=1}^{w/8} w_{m,n}\cdot\varphi_{m,n}^{k,l} + \sum_{c\in\mathcal{E}_{k,l}} w_c\cdot\psi_c^{k,l}. \end{cases} \tag{25}$$

In the revised Eq. (25), we introduce three types of weights, i.e., the mode weight $w_{k,l}$, the block weight $w_{m,n}$ and the pairwise weight $w_c$. The mode weight $w_{k,l}$ possesses a larger value on high-frequency modes than low ones, thus widening the difference between the corresponding energies of different frequency modes associated with the steganography detectability. As for the setting of $w_{k,l}$, we first scan the DCT modes with Zig-Zag order and then set the last $p$ DCT modes with $w_{k,l} = 5$ and $w_{k,l} = 1$ for the other ones. For QF = 75 and QF = 95, $p$ are 16 and 3, respectively. The block weight $w_{m,n}$ assigns smaller values to the unary potentials in the DCT blocks of more complex regions. To achieve this, the classic spatial steganographic scheme S-UNIWARD [16] is applied to obtain the costs $\boldsymbol{\rho}' = (\rho'^{i,j}_{m,n})^{h\times w}$ of the decompressed spatial image $\mathbf{Z}$, and then summed $\boldsymbol{\rho}'$ over blocks and make it as the block weight $w_{m,n}$, i.e.,

$$w_{m,n} = \sum_{i=0}^{7}\sum_{j=0}^{7} \rho'^{i,j}_{m,n}. \tag{26}$$

where $\rho'^{i,j}_{m,n}$ is the distortion cost of the $(i,j)$-th pixel in the $(m,n)$-th decompressed spatial block. As for the pairwise weight $w_c$, it is dedicated to pairwise potentials w.r.t. two adjacent DCT blocks,

which allows us to simply compute $w_c$ by merging the two involved block weights, e.g., $w_c = (w_{m,n} + w_{m,n+1})/2$.

To this end, the energy of the entire image $\mathbf{X}$ associated with the statistical detectability has been eventually determined so that we can obtain the optimal embedding probabilities $\boldsymbol{\beta}^*$ with the given payload constraint by minimizing the $\hat{E}(\boldsymbol{\beta} \mid \mathbf{X})$, corresponding to the minimum detectability, i.e.,

$$
\begin{cases}
\min_{\boldsymbol{\beta}} \hat{E}(\boldsymbol{\beta} \mid \mathbf{X})) = \sum_{k=0}^{7} \sum_{l=0}^{7} \hat{E}_{k,l}(\boldsymbol{\beta} \mid \mathbf{S}) \\
\text{s.t.} \sum_{k=0}^{7} \sum_{l=0}^{7} \sum_{m=1}^{h/8} \sum_{n=1}^{w/8} H\left(\beta_{m,n}^{k,l}\right) = L
\end{cases}, \quad (27)
$$

where $H(x)$ is the information entropy that $H(x) = -xlog(x) - (1-2x)log(1-2x)$, $L$ is the given embedding payload.

Notably, the optimization of Eq. (27) is convex and consequently has a global minimum since all its $E_{k,l}(\boldsymbol{\beta} \mid \mathbf{S})$ terms are convex functions regarding $\beta_{m,n}^{k,l}$. To deal with this optimization, the classical Lagrange multiplier method is employed on the Eq. (27), and have

$$
\frac{\partial}{\partial \beta_{m,n}^{k,l}} \left( F(\boldsymbol{\beta}) - \frac{1}{\gamma} \left( \Phi\left(\beta_{m,n}^{k,l}\right) - L \right) \right) = 0, \quad (28)
$$

where $\gamma > 0$ is the Lagrange multiplier. After some simple arithmetic, we can obtain:

$$
\gamma \left( \mathbf{U}_{m,n}^{k,l} + \mathbf{V}_{m,n}^{k,l} \right) = \frac{1}{\beta_{m,n}^{k,l}} \log \left( \frac{1}{\beta_{m,n}^{k,l}} - 2 \right), \quad (29)
$$

where

$$
\mathbf{U}_{m,n}^{k,l} = \frac{2}{(\sigma_{m,n}^{k,l})^4} + \sum_{(a,b)\in\mathcal{N}_{m,n}^{k,l}} \frac{2}{(\sigma_{m,n}^{k,l})^4(1-(\varrho_{m,n}^{a,b})^2)^2}, \quad (30)
$$

$$
\mathbf{V}_{m,n}^{k,l} = \sum_{(a,b)\in\mathcal{N}_{m,n}^{k,l}} \frac{2(\varrho_{m,n}^{a,b})^2}{(\sigma_{m,n}^{k,l})^2(\sigma_{a,b}^{k,l})^2(1-(\varrho_{m,n}^{a,b})^2)^2} \frac{\beta_{a,b}^{k,l}}{\beta_{m,n}^{k,l}}, \quad (31)
$$

and $\mathcal{N}_{m,n}^{k,l} = \{(m-1,n), (m+1,n), (m,n-1), (m,n+1)\}$ is the set of the indexes of the four-element cross-neighborhood of $x_{m,n}^{k,l}$, $(\sigma_{m,n}^{k,l})^2$ is the variance of $x_{m,n}^{k,l}$, $\varrho_{m,n}^{a,b}$ is the correlation coefficient between $x_{m,n}^{k,l}$ and its neighbor $x_{a,b}^{k,l}$. Once $(\sigma_{m,n}^{k,l})^2$ and $\varrho_{m,n}^{a,b}$ for each of the DCT coefficients are given, the embedding probabilities can be easily determined by numerically solving the equation (29).

To quickly solve Eq. (29) for all DCT coefficients, the inverse function to $f(x) = x \log(x-2)$ was tabulated, and a binary search is applied to find the corresponding Lagrange multiplier $\gamma$. Note that there is a quantity, i.e., $\beta_{a,b}^{k,l}/\beta_{m,n}^{k,l}$, in Eq. (31) that will hinder the rapid implementation of a lookup table in the solving process. To cope with this, an iterative optimization approach is developed for solving Eq. (29) until it converges. Specifically, in each new iteration, the quantity $\beta_{a,b}^{k,l}/\beta_{m,n}^{k,l}$ is firstly calculated by the $\boldsymbol{\beta}$ obtained in the last iteration. As for the first iteration, the quantity can be set to 1 for brevity. In addition, a trick is also used for fast solving the Eq. (29) that dynamically adjusts the search range of the Lagrange multiplier according to the results obtained in the last iteration. In this way, our scheme converges in only three iterations, and the $\boldsymbol{\beta}^*$ corresponding to the least detectability can be accordingly obtained.

Subsequently, it can be converted to the steganographic distortion cost by

$$
\rho_{m,n}^{k,l} = \log \left( 1/\beta_{m,n}^{k,l} - 2 \right). \quad (32)
$$

## 4 PARAMETER ESTIMATION

Recall Eq. (30) and Eq. (31), determining the $(\sigma_{m,n}^{k,l})^2$ and $\varrho_{m,n}^{a,b}$ is the key to solving Eq. (27). To estimate the variances of DCT coefficients, in J-MiPOD [3], Rémi et al. extended the pixel variance estimator proposed in MiPOD to JPEG images with effective modifications, achieving low computational complexity and respectable empirical security. Given these benefits, we adopt this improved estimator to compute the DCT coefficient variance $(\sigma_{m,n}^{k,l})^2$, which is briefly summarised as follows:

**step 1**: Decompress the JPEG image $\mathbf{X}$ to the corresponding spatial image $\mathbf{Z}$ using Eq. (2).

**step 2**: Subtract the estimated pixel expectations, which are obtained by using a Wiener filter with a window size $2 \times 2$, from $\mathbf{Z}$ to acquire image residuals.

**step 3**: Fit the local residuals of block size $p \times p$ via a linear parametric model with two-dimensional trigonometric polynomials of size $p^2 \times q$, and consequently obtain the estimated noise $\xi_{m,n}^{i,j}$ and pixel variance $(\zeta_{m,n}^{i,j})^2$ through the maximum likelihood estimation. We set set $p = 5$ and $q = 6$ in our scheme.

**step 4**: Compute the variance of each DCT coefficient by leveraging the linearity of DCT:

$$
(\sigma_{m,n}^{k,l})^2 = \sum_{i=0}^{7} \sum_{j=0}^{7} (f_{i,j}^{k,l})^2 (\zeta_{m,n}^{i,j})^2 / q_{k,l}^2. \quad (33)
$$

As for the estimation of the correlation coefficients between the DCT coefficients from the same mode over the adjacent blocks, we propose a simple yet efficient estimation method. Overall, we foremost compute a block-level correlation coefficient between two adjacent blocks and then assign this block-level estimation to the 64 pairs of DCT coefficients with the same mode in these two adjacent blocks to serve as their respective correlation coefficients. Formally, the block-level correlation coefficients are efficiently computed by applying the Pearson product-moment correlation coefficient formula to the estimated noises obtained in the variance estimation. To this end, the correlation coefficient $\varrho_{m,n}^{a,b}$ in (30) and (31) is formulated as:

$$
\varrho_{m,n}^{a,b} = \frac{\sum_{i=0}^{7} \sum_{j=0}^{7} \left( \xi_{m,n}^{i,j} - \overline{\xi_{m,n}^{i,j}} \right) \left( \xi_{a,b}^{i,j} - \overline{\xi_{a,b}^{i,j}} \right)}{\sqrt{\sum_{i=0}^{7} \sum_{j=0}^{7} \left( \xi_{m,n}^{i,j} - \overline{\xi_{m,n}^{i,j}} \right)^2} \sqrt{\sum_{i=0}^{7} \sum_{j=0}^{7} \left( \xi_{a,b}^{i,j} - \overline{\xi_{a,b}^{i,j}} \right)^2}}, \quad (34)
$$

where $\overline{\xi_{m,n}^{i,j}}$ and $\overline{\xi_{a,b}^{i,j}}$ are the mean values of the estimated noises in the $(m,n)$-th block and the $(a,b)$-th block, respectively. For numerical stability, we further set $(\sigma_{m,n}^{k,l})^2 = max((\sigma_{m,n}^{k,l})^2, 10^{-10})$ and $|\varrho_{m,n}^{a,b}| = min(|\varrho_{m,n}^{a,b}|, 0.99)$.

# 5 EXPERIMENTAL RESULTS AND ANALYSIS

## 5.1 Experimental Settings

*5.1.1 Image datasets.* In this paper, we adopt two databases: (1) *BOSSBase* v1.01 [1]. It consists of 10,000 grayscale images with size of 512×512. All images are compressed into the JPEG domain with quality factors (QF) of 75 and 95 separately to generate two cover datasets for experiments. (2) *ALASKAv2* [2]. It provides different datasets with various sizes, formats, and QFs. We selectively download and use two datasets with QF=75 and 95, both of which contain 80,000 grayscale JPEG images with the size of 512×512. For simplicity, we separately select 10,000 images from these two JPEG image datasets for experiments. Note that all images in ALASKAv2 have been processed randomly using different development pipelines, which are more diverse and realistic than BOSSBase.

*5.1.2 Steganographic schemes.* To evaluate the security performance of our proposed CRF, we choose three SOTA JPEG steganographic schemes for comparison, including two heuristic-based schemes J-UNIWARD [16] and UERD [13], and one model-based scheme J-MiPOD [3]. For simplicity, all the involved schemes are simulated are their distortion bound with payload $\alpha \in \{0.1, 0.2, 0.3, 0.4, 0.5\}$ bpnzac (bit per non-zero AC DCT coefficient).

*5.1.3 Steganalyzers.* Five SOTA steganalysis features, CC-JRM [19], DCTR [15], GFR [27] and SCA-GFR [4], are employed to comprehensively assess the security performance. The steganalyzers are trained as binary classifiers using the steganalysis features with the FLD (Fisher Linear Discriminant) ensemble [20]. Typically, half of the cover and stego images are used for the training ensemble classifier, and the remaining are used for testing. The ultimate security performance is quantified by detection error rate $\bar{P}_E$ averaged over ten times of classification testing, and larger $\bar{P}_E$ indicates better empirical security.

## 5.2 Comparison to Prior Arts

To verify the advantages of our proposed CRF, we compare the CRF with three SOTA competitors, i.e., J-MiPOD, J-UNIWARD and UERD, under various payloads and QFs in resisting the detection of CC-JRM, DCTR, GFR, and SCA-GFR. The corresponding security performance on BOSSBase is summarized in Table I. It shows that for all the payloads and QFs, our CRF consistently achieves the best empirical security among the involved tested schemes. In specific, for CC-JRM, although CRF is slightly superior to the J-UNIWARD, it can outperform J-MiPOD and UERD by a significant margin, especially UERD, which can achieve a performance gain of up to 4.27% and 6.40% at QF=75 and QF=95 under 0.5 bpnzac, respectively. For DCTR and GFR, which are dedicated to detecting the decompressed spatial embedding change after modifying the DCT coefficients, the CRF consistently outperforms all competitors by a clear margin and is particularly resistant to more advanced GFR detection. As for SCA-GFR, which is currently the most advanced steganalysis feature, our CRF still shows consistent superiority, especially at QF=95. A close look at the comparison between the non-independent CRF and independent J-MiPOD, for all the involved steganalysis features, our CRF can outperform the J-MiPOD across the board, confirming that taking into account the correlation between the DCT coefficients by CRF model does indeed improve JPEG steganographic

security. Figure 2 illustrates the comparison of embedding change probability $\beta$ for J-MiPOD and CRF at QF=95 under 0.4 bpnzac. Compared to J-MiPOD, our proposed CRF favors centralized embedding, where some large regions containing a few texture DCT blocks as well as the boundaries between smoothed and textured regions are hardly used, and the payload is more evenly spread over the image.

In addition, we further verify the generalization of our proposed CRF on ALASKAv2 datasets, the results are collected in Table II. Note that the images in ALASKAv2 are more realistic and much more diverse than those from BOSSBase, due to the much more complex, realistic, and randomized development processes as well as the larger set of cameras. Table II shows similar trends in that the proposed CRF consistently outperforms its competitors, especially under larger payloads, indicating that our CRF is indeed general across different datasets.

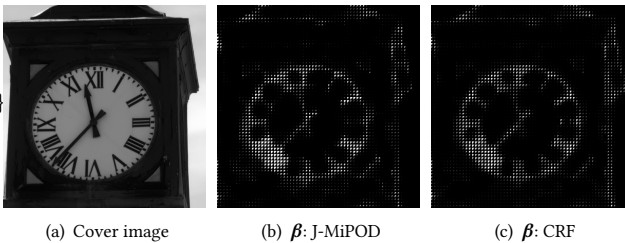

(a) Cover image (b) $\beta$: J-MiPOD (c) $\beta$: CRF

**Figure 2: (a) is a cover image, (b) and (c) are the embedding change probabilities of J-MiPOD and CRF, respectively, where the brighter points indicate larger probabilities.**

## 5.3 Time Complexity

For time complexity evaluation, we randomly selected 1000 512 × 512 images from BOSSBase to measure the average time-consuming in embedding distortion cost acquisition under QF=75 and QF=95 at 0.4 bpnzac. Not only the proposed CRF is evaluated, but also the other competitors are involved for comparison. The time complexity evaluation is performed with Matlab 2015b on a 3.20 GHz Intel CPU Xeon E-2836 with 64GB of memory running a 64-bit Windows 10 without parallel computing. As shown in Table 3, UERD owns the minimum time-consuming, and J-UNIWARD consumes far more time than the other three methods. Numerically, the time complexity of J-UNIWARD is nearly 180 times of UERD and about 3 times of CRF. Although the time complexity of our CRF is 3 times of J-MiPOD, it is still affordable.

## 6 CONCLUSION

In this paper, we propose a novel model-based JPEG steganographic scheme called CRF, wherein the Conditional Random Field (CRF) model with four-element cross-neighborhood is built to capture the mutual impact of embedding change probabilities by leveraging the correlation of DCT coefficients of the same mode in horizontally or vertically adjacent blocks. Overall, the CRF model is characterized by energy function defined as the weighted sum of a series of unary and pairwise potentials. Both kinds of potentials, in our

**Table 1: Average Detection Error $\bar{P}_E$ (in %) Comparison of the Proposed CRF with J-MiPOD, J-UNIWARD and UERD against Different Steganalyzers on BOSSBase Dataset with QF=75 and 95.**

| Feature | Method | Payload (bpnzac) for QF=75 | | | | | Payload (bpnzac) for QF=95 | | | | |
|---------|--------|------|------|------|------|------|------|------|------|------|------|
| | | 0.1 | 0.2 | 0.3 | 0.4 | 0.5 | 0.1 | 0.2 | 0.3 | 0.4 | 0.5 |
| CC-JRM | CRF | **47.03** | **41.33** | **34.84** | **27.78** | **21.01** | **49.71** | **48.12** | **45.34** | **41.07** | **35.65** |
| | J-MiPOD | 46.01 | 39.44 | 31.96 | 24.62 | 17.65 | 49.19 | 47.27 | 44.04 | 39.44 | 33.39 |
| | J-UNIWARD | 47.01 | 41.19 | 33.88 | 26.52 | 19.36 | 49.52 | 48.08 | 45.22 | 40.88 | 35.07 |
| | UERD | 45.87 | 38.81 | 31.12 | 23.43 | 16.74 | 49.13 | 46.56 | 41.87 | 35.83 | 29.25 |
| DCTR | CRF | **44.07** | **35.50** | **26.41** | **18.13** | **11.53** | **48.72** | **45.55** | **40.80** | **35.13** | **28.23** |
| | J-MiPOD | 42.02 | 31.57 | 21.22 | 13.16 | 7.55 | 48.17 | 44.12 | 38.76 | 32.56 | 25.52 |
| | J-UNIWARD | 43.85 | 34.24 | 24.15 | 15.44 | 9.04 | 48.70 | 45.44 | 40.02 | 33.50 | 26.21 |
| | UERD | 42.97 | 32.91 | 22.96 | 14.72 | 8.72 | 47.84 | 43.32 | 37.23 | 30.31 | 22.43 |
| GFR | CRF | **42.57** | **32.22** | **22.82** | **14.31** | **8.33** | **47.76** | **43.58** | **38.14** | **31.13** | **24.24** |
| | J-MiPOD | 41.23 | 29.64 | 19.27 | 11.18 | 6.36 | 47.01 | 41.38 | 35.15 | 27.73 | 20.45 |
| | J-UNIWARD | 41.07 | 28.64 | 18.18 | 10.22 | 5.56 | 47.49 | 42.65 | 35.63 | 27.84 | 20.19 |
| | UERD | 39.78 | 27.53 | 17.81 | 10.32 | 6.01 | 45.98 | 39.45 | 32.13 | 24.68 | 17.73 |
| SCA-GFR | CRF | **39.27** | **27.58** | **17.74** | **11.16** | **6.54** | **46.73** | **41.02** | **34.87** | **28.51** | **22.50** |
| | J-MiPOD | 38.51 | 26.19 | 16.64 | 10.14 | 5.85 | 45.56 | 38.76 | 32.21 | 25.46 | 19.54 |
| | J-UNIWARD | 35.94 | 23.25 | 14.15 | 8.03 | 4.45 | 46.23 | 40.32 | 33.64 | 26.70 | 20.41 |
| | UERD | 29.83 | 18.18 | 11.01 | 6.74 | 4.05 | 40.91 | 32.52 | 25.87 | 19.91 | 14.94 |

**Table 2: Average Detection Error $\bar{P}_E$ (in %) Comparison of the Proposed CRF with J-MiPOD, J-UNIWARD and UERD against Different Steganalyzers on ALASKAv2 Dataset with QF=75 and 95.**

| Feature | Method | Payload (bpnzac) for QF=75 | | | | | Payload (bpnzac) for QF=95 | | | | |
|---------|--------|------|------|------|------|------|------|------|------|------|------|
| | | 0.1 | 0.2 | 0.3 | 0.4 | 0.5 | 0.1 | 0.2 | 0.3 | 0.4 | 0.5 |
| CC-JRM | CRF | **46.50** | **40.48** | **33.79** | **27.18** | **20.45** | 47.30 | 42.07 | 36.24 | 30.48 | 24.60 |
| | J-MiPOD | 45.81 | 39.36 | 32.15 | 24.85 | 18.55 | 46.80 | 41.14 | 35.08 | 28.97 | 23.06 |
| | J-UNIWARD | 46.11 | 39.31 | 32.37 | 25.10 | 18.90 | 46.36 | 40.65 | 34.81 | 28.11 | 22.64 |
| | UERD | 45.84 | 39.22 | 31.53 | 24.75 | 17.88 | **47.43** | **42.54** | **37.53** | **32.16** | **27.05** |
| DCTR | CRF | **43.97** | **35.12** | **26.02** | **18.41** | **12.53** | **48.30** | **44.55** | **39.17** | **33.82** | **27.70** |
| | J-MiPOD | 42.58 | 32.69 | 23.23 | 15.62 | 10.62 | 47.82 | 43.42 | 38.52 | 32.50 | 25.89 |
| | J-UNIWARD | 43.08 | 33.41 | 24.39 | 16.96 | 11.27 | 48.02 | 43.69 | 38.24 | 31.71 | 25.72 |
| | UERD | 42.75 | 32.87 | 23.36 | 16.02 | 10.39 | 48.05 | 44.03 | 38.74 | 33.26 | 26.58 |
| GFR | CRF | **43.72** | **35.35** | **26.52** | **18.91** | **13.14** | **48.45** | **45.62** | **41.60** | **36.81** | **31.61** |
| | J-MiPOD | 43.03 | 33.99 | 24.53 | 17.08 | 11.64 | 48.37 | 44.87 | 40.96 | 35.90 | 30.15 |
| | J-UNIWARD | 41.58 | 30.87 | 21.22 | 13.83 | 8.66 | 48.28 | 44.55 | 39.75 | 34.01 | 28.12 |
| | UERD | 41.70 | 31.77 | 22.44 | 15.37 | 10.31 | 47.71 | 43.42 | 38.48 | 33.04 | 27.02 |
| SCA-GFR | CRF | **42.17** | **32.36** | **23.72** | **16.48** | **11.14** | **48.06** | **44.26** | **39.85** | **35.02** | **30.16** |
| | J-MiPOD | 42.01 | 32.04 | 23.06 | 15.58 | 10.57 | 47.85 | 43.99 | 39.57 | 34.49 | 29.12 |
| | J-UNIWARD | 39.34 | 28.10 | 19.08 | 12.43 | 7.89 | 47.84 | 43.72 | 39.05 | 33.71 | 28.26 |
| | UERD | 35.86 | 25.57 | 17.52 | 12.06 | 8.07 | 45.66 | 40.29 | 35.10 | 29.83 | 25.19 |

**Table 3: Average Time-Consuming (in Seconds) in Embedding Distortion Cost Acquisition on Randomly Selected 1000 Images from BOSSBase under QF=75 and QF=95 at 0.4 bpnzac.**

| QF | UERD | J-UNIWARD | J-MiPOD | CRF(CPU) |
|----|------|-----------|---------|----------|
| 75 | 0.0401 | 6.7765 | 0.6061 | 1.0441 |
| 95 | 0.0425 | 6.7527 | 0.5739 | 1.0608 |

scheme, are formulated as the KL divergence between the statistical distributions of cover and stego. Following this way, secure JPEG steganography is then formulated as the optimization problem of minimizing that energy function. The optimal embedding change probabilities corresponding to the least detectability can be eventually obtained by optimizing the constructed energy function with the given payload constraint after some necessary parameter estimation, based on which, the steganographic distortion cost can be converted accordingly. Numerous results show that the proposed CRF can not only surpass the SOAT heuristic-based UERD and J-UNIWARD, but also outperform currently the most advanced model-based J-MiPOD by a clear margin with affordable time complexity.

In the current work, we built a CRF model to capture the mutual impact of embedding change probabilities, in the future, we will go a step further to directly capture the mutual impact of embedding changes to further improve JPEG steganography, which will lead us to investigate the model-based asymmetric steganography embedding.

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
