# OpenReview forum: "Model-Based Non-Independent Distortion Cost Design for Effective JPEG Steganography"
_acmmm.org/ACMMM/2024/Conference — MM2024 Poster_

### Official Review · Reviewer_1kzY · 2024-05-21

**Rating:** 1
**Confidence:** 3

**Summary:**

This paper covers an interesting topic about model-based JPEG image steganography. It proposes a novel steganographic scheme utilizing the Conditional Random Field (CRF) model with the four-element cross-neighborhood to capture the dependencies among DCT coefficients for JPEG steganography with symmetric embedding.

**Strengths:**

The techniques used appear to be correct. Moreover, this paper is well-written, clear, and easy to understand.

**Limitations:**

1) The innovation is limited, the pairwise conditional random field model used in this paper already has the related method [1] in JPEG domain steganography.
2) There is a lack of comparison with some of the most advanced model-based JPEG steganography methods, such as references [1, 2].
3) The steganalyzers used in the security experiments are all handcrafted-based. It is suggested to add CNN-based steganalyzers.
4) It is interesting to see how the hiding affects the different DCT frequency modes.
5) It is suggested to evaluate comprehensively the complexity of the algorithm through multiple indicators.

[1] Pan et al., Efficient JPEG Image Steganography Using Pairwise Conditional Random Field Model. SP 2024.

[2] Li et al., Quaternary Quantized Gaussian Modulation With Optimal Polarity Map Selection for JPEG Steganography, IEEE TIFS, 2023.

**Suitability:**

2

---

### Official Review · Reviewer_NFFc · 2024-05-23

**Rating:** 2
**Confidence:** 4

**Summary:**

This paper proposes a model-based steganographic scheme using the conditional random field, which uses four-element cross-neighborhood to capture the dependencies among DCT coefficients. The conditional random field model is designed by a energy function, which is defined as the weighted sum of a series of unary and pairwise potentials. Experiments show the effectiveness of the proposed steganography, compared with previous methods.

The proposed method demonstrates superior performance compared to the baseline methods. However, the contribution is limited, and the baselines and steganalysis methods are obsolete.

**Strengths:**

1) The paper in general is well written and well organized.

2) The paper is technically sound, and the method description is succinct yet clear.

**Limitations:**

1) Lack of significant contribution. The contributions of this paper are marginal. The work presented is merely an improvement over existing literature, without introducing substantial novel insights or breakthroughs.

2) Outdated comparative methods. The comparative methods used in this paper are outdated, primarily relying on techniques that are over a decade old. The paper should compare its results with more recent methods, especially the methods with cost designs based on learning mechanisms, to provide a more fair comparison.

3) Outdated steganalysis methods. The steganalysis techniques employed in this paper are also quite dated. The authors should utilize the latest deep learning-based steganalysis methods to assess the steganographic security, which would offer a more accurate and contemporary evaluation.

4) The visual comparison between Figure 2 (b) and (c) is not clear. The claimed differences related to steganographic embedding are not visually evident. The authors need to provide a clearer illustration of these differences to support their claims effectively.

5) The theoretical analysis presented in the paper is somewhat overly complicated and could benefit from simplification.

**Suitability:**

2

---

### Official Review · Reviewer_AhLK · 2024-05-24

**Rating:** 5
**Confidence:** 3

**Summary:**

This paper proposes a novel model-based steganographic scheme for JPEG images when the side-information of images is unavailable. It leverages a Conditional Random Field (CRF) model to capture dependencies among Discrete Cosine Transform (DCT) coefficients. This model aims to enhance security in JPEG steganography by minimizing the Kullback-Leibler (KL) divergence between the statistical distributions of cover and stego images. The proposed scheme, characterized by its delicately designed energy function, outperforms previous methods, including the independent J-MiPOD, in empirical evaluations.

**Strengths:**

Highly explainable: The proposed method is based on complete mathematical derivations. By defining the energy function as a weighted sum of unary and pairwise potentials, and optimizing it under a payload constraint, the authors ensure a robust theoretical foundation for the proposed scheme​​.
Addresses real-world constraints: The scheme is designed to work without any side information of images, making it applicable to situations with only a single compressed JPEG image. This enhances its practicality in a wider range of real-world scenarios.
Superior Performance: The experimental results indicate that the proposed method consistently outperforms existing methods and achieves lower detection error rates across four steganalysis features (CC-JRM, DCTR, GFR, SCA-GFR), especially under higher payload conditions.

**Limitations:**

Computational Complexity: The CRF model, despite its superior performance, has a higher computational complexity compared to some existing methods like UERD and J-MiPOD. Although the paper mentions that the time complexity is affordable, it is still significantly higher than other methods, which could limit its practicality in real-time applications​​.
Limited Scope of Embedding Strategies: The paper focuses solely on ternary symmetric embedding. This limitation might restrict the applicability of the proposed method to other embedding strategies or more complex steganographic scenarios. Moreover, This paper only consider the correlations of DCT coefficients from the same mode in horizontally or vertically adjacent blocks. Does such a consideration have certain limitations?
Limited comparative models: To evaluate the security performance of CRF, the authors chose two heuristic-based schemes and one model-based scheme. However, they did not use the current state-of-the-art deep learning-based steganography algorithms for comparison.

**Suitability:**

2

---

### Official Review · Reviewer_MuSm · 2024-05-24

**Rating:** 6
**Confidence:** 4

**Summary:**

This paper introduces a novel model-based steganographic scheme to improve the security of JPEG steganography. Using the pairwise Conditional Random Field (CRF) model with a four-element cross-neighborhood, the proposed method effectively captures the dependencies among DCT coefficients. Extensive experimental results validate the effectiveness of the proposed scheme, which significantly outperforms the previous methods, especially J-MiPOD.

**Strengths:**

The State-of-the-Art section provides a detailed introduction to distortion cost functions, highlighting their tutorial implications.

Figure 1. clearly describes the DCT modes splitting method, offering a visual and explanatory aid to understand the process.

Comprehensive experiments have been conducted to thoroughly demonstrate the superiority of the CRF method.

**Limitations:**

The validity of the claim that CRF's time complexity is affordable is uncertain, as there is no clarification on what constitutes affordable.

Consider providing the rationale for why QF of 75 and 95 were chosen for evaluating performance gains, as this can further enhance the interpretation of the results.

Consider incorporating additional image comparisons alongside Figure 2., this would allow for a more comprehensive assessment of embedding change probability for J-MiPOD and CRF.

**Suitability:**

2

---

### Meta-Review · Area_Chair_y4yz · 2024-07-03

**Recommendation:** Accept (Poster)
**Confidence:** 5

**Metareview:**

As pointed out by two reviewers, the novelty of this work is rather limited. Many of the proposed techniques are slightly improved version of the existing ones, making the technical contributions marginal. These aspects should be addressed adequately in the CR version. Given the review comments and the final review ratings, I recommend paper acceptance.